# Zonulin-Dependent Intestinal Permeability in Children Diagnosed with Mental Disorders: A Systematic Review and Meta-Analysis

**DOI:** 10.3390/nu12071982

**Published:** 2020-07-03

**Authors:** Birna Asbjornsdottir, Heiddis Snorradottir, Edda Andresdottir, Alessio Fasano, Bertrand Lauth, Larus S. Gudmundsson, Magnus Gottfredsson, Thorhallur Ingi Halldorsson, Bryndis Eva Birgisdottir

**Affiliations:** 1Unit for Nutrition Research, Faculty of Food and Nutritional Sciences, University of Iceland, 101 Reykjavik, Iceland; hes35@hi.is (H.S.); eda11@hi.is (E.A.); tih@hi.is (T.I.H.); beb@hi.is (B.E.B.); 2Faculty of Medicine, School of Health Sciences, University of Iceland, 101 Reykjavik, Iceland; bertrand@landspitali.is (B.L.); magnusgo@landspitali.is (M.G.); 3Department of Child and Adolescent Psychiatry, Landspitali University Hospital, 101 Reykjavik, Iceland; 4Division of Pediatric Gastroenterology and Nutrition, Harvard Medical School, MassGeneral Hospital for Children, Boston, MA 02114, USA; afasano@mgh.harvard.edu; 5Faculty of Pharmaceutical Sciences, School of Health Sciences, University of Iceland, 101 Reykjavik, Iceland; larussg@hi.is; 6Department of Scientific Affairs, Landspitali University Hospital, 101 Reykjavik, Iceland

**Keywords:** children, adolescents, intestinal permeability, mental disorders, zonulin, haptoglobin, attention deficit and hyperactivity disorder (ADHD), autism spectrum disorder (ASD), obsessive–compulsive disorder (OCD), systematic review, meta-analysis

## Abstract

Worldwide, up to 20% of children and adolescents experience mental disorders, which are the leading cause of disability in young people. Research shows that serum zonulin levels are associated with increased intestinal permeability (IP), affecting neural, hormonal, and immunological pathways. This systematic review and meta-analysis aimed to summarize evidence from observational studies on IP in children diagnosed with mental disorders. The review follows the Preferred Reporting Items for Systematic Reviews and Meta-Analyses (PRISMA) guidelines. A systematic search of the Cochrane Library, PsycINFO, PubMed, and the Web of Science identified 833 records. Only non-intervention (i.e., observational) studies in children (<18 years) diagnosed with mental disorders, including a relevant marker of intestinal permeability, were included. Five studies were selected, with the risk of bias assessed according to the Newcastle–Ottawa scale (NOS). Four articles were identified as strong and one as moderate, representing altogether 402 participants providing evidence on IP in children diagnosed with attention deficit and hyperactivity disorder (ADHD), autism spectrum disorder (ASD), and obsessive–compulsive disorder (OCD). In ADHD, elevated serum zonulin levels were associated with impaired social functioning compared to controls. Children with ASD may be predisposed to impair intestinal barrier function, which may contribute to their symptoms and clinical outcome compared to controls. Children with ASD, who experience gastro-intestinal (GI) symptoms, seem to have an imbalance in their immune response. However, in children with OCD, serum zonulin levels were not significantly different compared to controls, but serum claudin-5, a transmembrane tight-junction protein, was significantly higher. A meta-analysis of mean zonulin plasma levels of patients and control groups revealed a significant difference between groups (*p* = 0.001), including the four studies evaluating the full spectrum of the zonulin peptide family. Therefore, further studies are required to better understand the complex role of barrier function, i.e., intestinal and blood–brain barrier, and of inflammation, to the pathophysiology in mental and neurodevelopmental disorders. This review was PROSPERO preregistered, (162208).

## 1. Introduction

Worldwide, up to 20% of children and adolescents experience mental, behavioral, and neurodevelopmental disorders, and these are the leading cause of disability in young people [1,2]. The etiology of most mental disorders, including behavioral and neurodevelopmental disorders, is unknown, but genetic factors, biochemical imbalances, and reactions to environmental stressors seem to play a role [3,4]. Known mental disorders in children and adolescents are, e.g., ADHD, anxiety disorders, OCD, ASD, depressive disorders, bipolar disorder (BD) and schizophrenia, and co-morbidity is frequently seen. Characteristics of mental disorders such as ADHD, ASD, and OCD are partly overlapping, i.e., poor academic performance, impaired social skills, hyperactivity, and impulsive behavior. The interplay between mental disorders, IP, and the intestinal microbiota has become a rapidly growing area of research. Increased IP has been examined with regard to emotional, behavioral, and neurodevelopmental disorders such as ADHD, ASD, major depressive disorders (MDD), obsessive–compulsive disorder (OCD), and schizophrenia [5,6,7,8,9,10,11,12,13,14].

The digestive tract is the body’s largest interface with the external environment. The function of the intestines is to be selectively permeable by allowing absorption of needed nutrients from the lumen, through the intestinal cells, into the intestinal milieu and the circulation as well as preventing the transfer of harmful entities by barrier function [15]. Inherent to the rationale of a gastro-intestinal tract in the pathogenesis of mental disorders is the model of IP of the intestinal barrier [16,17].

Zonulin is a family of structurally and functionally related peptides whose archetype member is pre-haptoglobin-2. Zonulin family peptides (from now on referred to as zonulin) are the only known physiologic modulator of intercellular tight junctions described so far and affect mechanisms that regulate the intestinal epithelial paracellular pathway [18,19,20]. Zonulin seems to be the primary modulator for regulating permeability in both gut–blood and blood–brain barriers and has been used as a clinical indicator of intestinal permeability [17]. Zonulin is also a potential inflammatory marker and contributes to intestinal innate immunity. Increased IP coinciding with inflammation has been described in mental disorders in adults [21,22,23,24,25,26]. Zonulin has been associated with low-grade inflammation and autoimmune diseases, as well as ASD, which might have an autoimmune component [6,27,28]. Children with ASD have been found to display an increased immune reactivity against proteins such as gluten, found in certain types of grain, potentially as a result of increased IP [7,29,30]. Furthermore, in ASD, higher serum zonulin has been associated with social impairment compared to controls [5].

As in numerous gastro-intestinal (GI) disorders, chronic inflammation has been found in mental disorders [31]. Loss of the intestinal barrier function leads to an increased influx of dietary and microbial factors and may contribute to the progression of chronic inflammatory disorders [32]. Recent studies show that inflammation seems to play a role in the pathogenesis of depression, at least for a subset of individuals [33]. The pediatric literature also demonstrates mutually influencing bidirectional pathways between inflammation and depression [34]. Children with higher levels of interleukin 6 (IL-6) and C-reactive protein (CRP) are, for example, more likely to be depressed later in life [35].

Furthermore, type 2 tissue transglutaminase (TG2), an autoantigen in celiac disease, has been associated with various diseases in humans, i.e., inflammation, cancer, fibrosis, cardiovascular disease, and neurodegenerative diseases [36]. Moreover, the presence of antibodies against gluten has been described in gluten sensitivity, a common complaint in subgroups of ASD subjects [7]. However, transglutaminase 6 (TG6) is sensitive and specific in gluten ataxia. This extraintestinal manifestation can cause neurologic disorders, i.e., cerebellar gluten ataxia and peripheral neuropathy, and has been reported in schizophrenia, cerebral palsy, and gluten neuropathy [37]. The role of inflammation in the development of mental disorders needs further investigation.

The commensal microorganisms living within the gut lumen, i.e., the intestinal microbiota, can affect both the host’s physiology and pathology [38,39]. Recent studies reveal that an imbalance of the intestinal microbiota, i.e., intestinal dysbiosis [40], can lead to loss of barrier function and increase IP and is thought to be one of the mechanisms by which intestinal microbiota may affect the central nervous system (CNS).

Recently, a similar permeability mechanism related to the CNS and the blood–brain barrier has been described [11]. Claudin-5, a transmembrane tight-junction protein, is expressed in the blood–brain barrier and plays a crucial role in maintaining the integrity of the brain endothelial barrier, and changes in this function may contribute to enhanced permeability of the endothelial barrier of the brain [9,11]. Increased expression of claudin-5 has been associated with mental disorders. Serum claudin-5 levels have been found significantly higher in OCD subjects [9]. Furthermore, an aberrant pattern of claudin-5 immunoreactivity has been observed in schizophrenia subjects [10].

Gut–brain axis (GBA) involves bidirectional communication between the brain and the intestinal tract, linking the enteric and central nervous systems, which affects mental health, impacting both mood and behavior [41]. The GBA is considered the pathway that explains how the gut influences mood, cognition, and mental health, as well as how the brain influences intestinal functions such as the activity of functional immune effector cells [42,43]. Research has revealed alteration of GBA in ADHD [44]. Clinical data indicate that psychological stress evokes a weaker activation of the hypothalamic–pituitary–adrenal axis (HPA) in children with ADHD compared to the control group [45].

Research shows that children with ASD display abnormal gut permeability as measure by the lactulose/mannitol ratio test when compared to controls [46]. Furthermore, higher serum zonulin has been associated with social impairment compared to controls [6], indicating that children with mental disorders are more prone to increased IP. A better understanding of the relationship between the GBA and the neuro-immune system provides a novel approach for the management of mental disorders. There is an urgent need to investigate these emerging interrelations further. To the authors’ knowledge, there is no existing systematic review on the association between IP and mental disorders in children. 

The aim of this systematic review was to summarize evidence from observational studies on IP, evaluating intrinsic biomarkers or endotoxins as exposure in children and adolescents diagnosed with mental disorders.

## 2. Materials and Methods

This systematic review was conducted and reported according to the Preferred Reporting Items for Systematic Reviews and Meta-Analysis (PRISMA) guidelines [47] (PRISMA checklist in Appendix B) to improve both study accuracy and transparency. Relevant experts from different health disciplines joined the review team for additional quality management, such as for search strategy design, clinical research, and biostatistics. Every attempt was made to develop a comprehensive protocol with appropriate literature searches, screening, data extraction, and reporting methodologies, along with detailed eligibility criteria. The review was preregistered with PROSPERO (162208). Any amendments to the protocol were dated and detailed rationale for the changes documented in PROSPERO.

### 2.1. Search Strategy

Selected bibliographic databases for observational studies were searched for this review. The bibliographic databases include PubMed, Cochrane CENTRAL, PsycINFO, and Web of Science. Searches were limited to observational study type publications with no language restrictions. A Grey literature search was performed. A manual search of key journals was performed online. Experts in the field were contacted for relevant literature on the topic under review. The last search was conducted in April 2020 but repeated before publication, retrieving one new study which had been published during the review process. Prior to the main search, the PROSPERO was searched to prevent the repetition of the review in question. To perform the best possible search for the review, combinations of free-text and subject headings (for example, Medical Subject Headings [MeSH] in PubMed) were used. The peculiarities of every database and consequences for the search strategy and the search strings were considered. Identification of controlled vocabulary terms for each database was attempted by retrieving studies from the database meeting the inclusion criteria for the review, and common text words were noted and the subject terms that the indexers have applied to the studies. The search was conducted using combinations of the search terms, and a detailed search string was developed. A set of search terms was designed according to the PICOs criteria. All terms within each concept of PICO were combined with the Boolean operator “OR” to increase the sensitivity of the search as it expands the retrieval (detailed sample search string in Appendix C). Search terms were identified by trying out ClinicalTrials.gov, amongst other things. Unless otherwise stated, search terms were free-text terms. Details of the date of the last search and database searched were also documented in addition to the period searched and an outline of the initial search results. 

### 2.2. Eligibility Criteria

Included studies were limited to human observational studies. In identifying the review question and developing the eligibility criteria, PICO was used as a guide [48]. Population studies on children and adolescents (0–18 years), diagnosed with one or more of the following according to validated criteria; anxiety disorders (AD), ASD, attention deficit disorder (ADD), ADHD, bipolar disorder (BD), major depressive disorder (MDD), manic-depressive disorder, OCD, and schizophrenia (SCZ). Mental disorders, diagnosed by a standard procedure such as experts’ diagnoses involving dimensional and multi-informant assessments, including standardized diagnostic instruments and validated rating scales, were eligible for inclusion. In addition, the study needed to address intestinal permeability. The exposures were biomarkers (intrinsic or endotoxins) for intestinal permeability such as zonulin, diamine oxidase (DAO), lipopolysaccharide (LPS), and lipopolysaccharide binding protein (LBP) in individuals diagnosed with one or more of the mental illnesses according to the validated criteria. Comparators were the same biomarkers (intrinsic or endotoxins) for intestinal permeability (Table 1).

### 2.3. Data Management

A study flow diagram was developed based on the PRISMA statement and flow diagram principle [47]. This was done in order to illustrate the results of the search/research and the process by which the records were screened and selected for the review. Results from the search were imported directly into the reference management system EndNote. All studies were screened for relevance using the full citation, abstract, and indexing terms before excluding studies as not relevant. Duplicates were removed, and the most recent and complete version of the studies was reviewed. All relevant studies were assessed for eligibility by three reviewers, B.A., as well as E.A. and H.S., splitting their part in two, according to the pre-specified inclusion and exclusion criteria.

### 2.4. Selection and Data Collection Process

The data was extracted by the reviewers using a predefined data extraction form (Appendix D). The data extracted included specific details about the patients, exposures, setting, and outcomes based on PICO. For missing information from the included studies, published reports of the individual studies were accessed, and authors were contacted for clarification or further information [49]. Studies, including relevant data/information but not fulfilling inclusion criteria were retained for later use, e.g., in the introduction/discussion chapters. The predefined data extraction form used to collect relevant information from each study included (but was not limited to) the following: Reviewer ID, date of extraction, general information (i.e., author details, title, citation, year of publication, source of funding/conflict of interest, doi/url), study characteristics (i.e., objective, limitations, inclusion-/exclusion criteria, types of participants, diagnosis), population (i.e., eligibility criteria, age, male/female proportion, prior treatment history), participant characteristics (i.e., age, gender, disease characteristics, diagnostic criteria), exposure/comparison and setting (i.e., biomarkers, comparator (unclear or none), number and type of clinical settings), outcome data/results (i.e., main outcome and effects addressed in the study, changes in the intestinal permeability, metabolic parameters and biomarkers, total number of participants included in the study), statistical techniques used, number of participants involved in analysis, and summary outcome data (i.e., dichotomous, continuous, estimate of) and outcome measures. 

### 2.5. Quality Assessment and Risk of Bias Tool 

All included articles were checked against the Strengthening the Reporting of Observational Studies in Epidemiology (STROBE) guideline by the lead researcher (BA). The STROBE was not applied for rating the studies but rather to investigate the author’s adherence to the guidelines in presenting their work, as the guidelines are not to act as a validation tool for the conducted study [50].

The risk of bias was assessed by the first author (B.A) as well as other reviewers (E.A., H.S.), according to the Newcastle–Ottawa Scale (NOS) assessing selection, comparability, exposure, and outcome and rated according to the criteria. This was based on the global evaluation of the study and informed by the NOS criteria [51]. The full NOS criteria are illustrated in Appendix E. 

### 2.6. Meta-Analysis

Quantitative synthesis was undertaken on studies including the same exposure and outcome, and mean and standard deviation were pooled, taking into account both within-study and between-study variability. For the zonulin data, a fixed-effects model was applied, using Cohens-d to calculate the effect size.

## 3. Results

### 3.1. Identification and Description of the Studies

The electronic search of PubMed, Cochrane CENTRAL, PsycINFO, and Web of Science identified 833 records. No references were identified in reference searching. Forty-four records were removed as duplicate leaving 789 remaining. Altogether, 776 were excluded from the title and abstract screening, leaving 13 records. Nine studies were excluded with reasons (Table 2), while the remaining four studies were included in the systematic review. One additional study was retrieved through a second search performed in May 2020, before publication and was added to the review. Out of the five included studies, four were included in a meta-analysis, since in the fifth one, only one peptide of the zonulin peptides family was evaluated. The study flow diagram, according to the PRISMA guidelines [47], showing the flow of the articles through the selection process, is illustrated in Figure 1. 

### 3.2. Description of Excluded Studies

The description of the excluded studies and reasons for exclusions are given in Table 2. The most common reason for exclusion was not fulfilling the inclusion criteria for the review, e.g., not clearly defined PICO (population, exposure, comparison, outcome) criteria or one of the PICOs criteria not fulfilling the inclusion criteria for the current systematic review. There were relatively few studies on children evaluating intrinsic biomarkers or endotoxins as exposure; the more commonly used method was lactulose/mannitol test, resulting in only 14 studies eligible for full-text review.

### 3.3. Summary of Main Results and Description of the Studies

For the pre-specified objective for this systematic review, there were five studies eligible, i.e., five case-control studies [5,6,7,8,9], with all patients diagnosed with mental disorders, compared to healthy controls and with the majority of recruited participants being males. The full characteristics of the included studies are shown in Table 3. One of the five studies included investigated the association between intestinal barrier function, immune response, and ADHD [5]; three investigated the association between impaired IP, immune responses, and serum zonulin/plasma haptoglobin levels and ASD [6,7,8]; and one investigated serum zonulin and claudin-5 levels in OCD [9]. 

Summary of main findings are listed in Table 4, and a summary of population characteristics and summary of participant’s characteristics are listed in Appendix A, respectively. Metabolic parameters and biomarkers are listed in Appendix A. Four [5,6,7,9] out of five studies analyzed only blood samples, and one study [8] analyzed both blood and stool samples. Özyurt et al. [5] compared 40 subjects diagnosed with ADHD to 41 healthy controls. Esnafoglu et al. [6] compared 32 subjects diagnosed with ASD to 33 healthy controls. Józefczuk et al. [7] compared 75 subjects diagnosed with ASD to 46 healthy controls. Rose et al. [8] compared 46 subjects diagnosed with ASD (with or without current or previous GI symptoms) to 41 healthy controls (with or without current or previous GI symptoms). Işık et al. [9] compared 24 subjects diagnosed with OCD to 24 healthy controls. The mean age of the included participants diagnosed with ASD and ADHD was seven and eight years, respectively, and 14 years in patients diagnosed with OCD Four studies [5,6,7,9] analyzed serum zonulin, and one study [8] analyzed plasma haptoglobin levels and was, therefore, not included in the meta-analysis.

No studies on the following mental disorders fulfilled the inclusion criteria for the current systematic review: anxiety disorders, attention deficit disorder, bipolar disorder, manic-depressive disorder, major depressive disorder, or schizophrenia. The association between IP and mental disorders was investigated by serum zonulin and plasma haptoglobin, in a total of 402 participants (217 patients and 185 controls);
ADHD:∘“Increased zonulin is associated with hyperactivity and social dysfunctions in children with attention deficit hyperactivity disorder” [5].ASD:∘“Increased serum zonulin levels as an intestinal permeability marker in autistic subjects” [6].∘“The occurrence of antibodies against gluten in children with autism spectrum disorders does not correlate with serological markers of impaired intestinal permeability” [7].∘“Differential immune responses and microbiota profiles in children with autism spectrum disorders and co-morbid gastro-intestinal symptoms” [8].OCD:∘“Serum zonulin and claudin-5 levels in children with obsessive–compulsive disorder” [9].

Serum zonulin was assessed in four studies [5,6,7,9], including 171 patients and 144 controls, and they are listed in Table 5. One study, Rose et al. [8], only reported plasma haptoglobin levels but not serum zonulin levels, which was confirmed through author contact (plasma haptoglobin levels are listed in Appendix A) and therefore not included in the meta-analysis. 

### 3.4. Study Quality 

Using the quality assessment tool, four articles were identified as strong and one as moderate. The risk of bias assessment of the included studies is summarized in Table 6.

All the studies got a full quality score for six of the nine parts of the quality assessment. The reason for not receiving a full quality score for selection of controls was that the controls were recruited in a hospital setting, or recruiting was not explained at all. To be eligible for a point for non-response rate, the same rate of exposure had to be mentioned for both cases and controls.

### 3.5. Meta-Analysis

Data were synthesized based on the outcome measure, i.e., biomarkers of intestinal permeability equating ADHD, ASD, and OCD, as mental disorders. Information on zonulin levels in cases and healthy controls (mean and standard deviation) was extracted from the four publications that reported such numbers [5,6,7,9]. Meta-analysis was conducted using Meta-Essential [61]. Study findings were combined using a fixed-effect model, and the effect size was expressed as Cohen’s d, effect size small-medium (*d* = 0.3–0.4) (Figure 2). Based on these four studies, the statistical heterogeneity was estimated as high evaluated using a fixed-effect model (*I*^2^ 71.52%). However, as the number of studies is only four, the true underlying heterogeneity is difficult to assess accurately [62]. Based on the Egger test for funnel asymmetry, no indication of publication bias was observed (*p* = 0.72). The combined estimate from the studies presented in Table 5 showed a statistically significant difference (*p* = *0*.001) between zonulin levels among cases (171) compared to controls (144). In one of the studies included in the pooled estimate, Józefczuk et al., [7] the measured zonulin levels were much lower compared to the other three studies (levels in cases 17.2 ± 15.7 ng/mL and 15.3 ± 5.9 ng/mL in healthy controls, *p* > 0.05). 

## 4. Discussion

The aim of this systematic review was to summarize the evidence on IP in children diagnosed with mental disorders. The main findings demonstrate that there is an association between increased plasma zonulin levels and impaired intestinal barrier function in ADHD and ASD but not as much in OCD. However, serum plasma levels of claudin-5, a tight-junction protein in the blood–brain barrier, were found to have severely increased in OCD patients. Performing the meta-analysis, on the four studies using the mean zonulin plasma levels of patients and control groups from four [5,6,7,9], revealed a significant difference between the groups (*p* = 0.001). However, Józefczuk et al. [7] reported very low mean serum zonulin levels (17.2 ± 15.7 ng/mL in cases and 15.3 ± 5.9 ng/mL in healthy controls) in both patients and control groups which may be explained by possible issues with their assessment kit in terms of sensitivity [63]. The fifth study included in the systematic review, Rose et al. [8], was excluded in the meta-analysis as they did not report findings on serum zonulin levels but plasma haptoglobin levels. 

The inclusion criteria for this systematic review were strict, and studies were only included if patients were diagnosed with mental disorders according to a standard diagnostic procedure such as experts’ diagnoses involving dimensional and multi-informant assessments, including recognized and validated rating scales. The quality of the included studies was rated strong [5,6,8,9] and moderate [7], respectively. However, none of the authors reported following the STROBE guideline in presenting their work.

The five included case-control studies assessed in total 402 children and adolescents, males (*n* = 186 cases), and females (*n* = 95 cases), but predominantly males, i.e., *n* = 186/95, respectively, excluding one study [8] as male/female proportion was not reported. Despite the differences between included studies, i.e., ADHD, ASD, and OCD, there was some overlap between ASDs and ADHD at the symptom level (social problems) as well as syndrome levels. Both ADHD and ASD have been linked to the atypical allocation of attentional resources, atypical performance monitoring, atypical processing of faces, and atypical early sensory processing in both visual and auditory domains. However, there are also atypical processing aspects that are disorder-specific [64,65,66,67]. 

Özyurt et al.’s [5] was the only study that fulfilled the inclusion criteria for IP And ADHD for the review, indicating few studies on the topic. There were significantly elevated levels of zonulin in ADHD compared to controls, and subjects with hyperactive/impulsive presentations had significantly elevated zonulin compared to children with other presentations. The level of zonulin was independently predicted by hyperactivity symptoms and Social Responsiveness Scale (SRS) scores in regression analysis, concluding that serum zonulin may be a biomarker for hyperactivity symptoms and social difficulties in ADHD [5]. Research shows that symptoms of inattention, hyperactivity, and impulsivity are all present in ADHD and can lead to severe problems in academic and social fields. Other symptoms are frequently seen, such as problems with social cues and difficulties in interpreting or respond appropriately to social situations [64]. 

Three studies fulfilled the inclusion criteria for the association of IP in ASD patients [6,7,8]. Esnafoglu et al. [6] identified a correlation between the Childhood Autistic Rating Scale (CARS) score, indicating the severity of autism, and the level of zonulin, but this was not observed in healthy control subjects. There was no correlation between age and zonulin levels. However, there were high zonulin levels detected in some healthy control group subjects indicating other factors contributing to the ASD pathogenesis. 

Józefczuk et al. [7] found a statistically significant negative relationship between age and concentration of zonulin in another study of IP in ASD. However, this group physiologically had an immature intestinal barrier because of the young age [68]. In previous research, GI symptoms have commonly been observed in patients with ASD. They are more common when compared to typically developing (TD) children, particularly altered bowel habits such as constipation and chronic abdominal pain, and this seems to correlate strongly with the severity of their ASD, suggesting these co-morbidities require attention [69,70,71,72]. When compared to TD children, children diagnosed with ASD are more likely to have at least one frequent GI symptom, which can be defined as gaseousness, constipation, painful bowel movement, diarrhea, or recurring abdominal pain. In addition, these subjects generally score worse on irritability, social withdrawal, stereotypy, and hyperactivity compared with children having no frequent GI symptoms [72].

Rose et al. [8] reported alterations in plasma haptoglobin levels in ASD subjects compared to controls as well as dysbiotic microbiota in fecal samples. Findings from this review suggest that children with ASD and GI symptoms have an imbalanced immune response resulting in a predisposition to impaired barrier function, which may contribute to their symptoms and clinical outcome [8]. Therefore, impaired intestinal barrier function may be a marker of ASD only in children with ASD and GI problems [8]. 

Işık et al.’s [9] was the only article that fulfilled the eligibility criteria for OCD and IP for this review. Concentrations of serum zonulin and claudin-5 levels were compared between OCD patients and healthy controls. The association between zonulin and claudin-5 concentrations and OCD severity were investigated, and serum claudin-5 levels were found to be significantly higher in OCD patients. However, serum zonulin levels were not significantly different between the groups, which could indicate different mechanisms compared to ADHD, ASD, as well as other types of mental disorders.

The nine excluded studies upon full-text review, not fulfilling the inclusion criteria for the review [52,53,54,55,56,57,58,59,60], included three types of mental disorders, i.e., major depressive disorder, schizophrenia, and ASD. All of the studies were in children, except one including children and adults up to 35 years [54]. The most common reason for exclusion was the use of lactulose/mannitol test and no assessment of intrinsic biomarkers nor exotoxins. However, the findings supported a mechanistic pathway linking the sympathetic nervous system activation to increased IP as involving activation of the immune system [52]. Moreover, findings from Delaney et al. [54] revealed a potential interrelated inflammatory contribution to the pathophysiology of psychosis. Iovene et al. [57] found low/mild intestinal inflammation as well as increased IP together with the presence of GI symptoms in ASD subjects compared to controls, and Jyonouchi et al. [58] found innate immune response against LPS in ASD subjects. Lau et al. [59] reported that ASD children had significantly higher levels of IgG antibody to gliadin compared with unrelated healthy controls. They concluded increased GI symptoms pointed to mechanisms involving immunologic abnormalities and increased IP in a subset of ASD children.

On the other hand, Dalton et al. [53] and Pusponegoro et al. [60] found no difference in IP between ASD children and controls. According to De Santis et al. [55], the comparison of average values of increased IP among groups, i.e., ASD and controls, did not produce statistically significant differences, and Gabriele et al. [56] did not report increased IP in ASD infants. Therefore, though not included in the current systematic review and meta-analysis, five of the studies, including children diagnosed with ASD, one research on major depressive disorder, and one study on schizophrenia, support the findings from this systematic review and meta-analysis.

Clinical and animal model studies of the association between intestinal barrier function and mental disorders have been reviewed, with the majority of reports on ASD and schizophrenia [46]. Research shows that children with ASD display abnormally high urinary excretion of the sugars used as permeability markers, i.e., in the lactulose/mannitol ratio test when compared to controls [46]. Data from 90 ASD children as well as 146 relatives showed that 36.7% of ASD patients displayed an abnormal intestinal permeability (lactulose/mannitol ratio) when compared with 21.2% of their first-degree relatives (versus 4.8% of adult controls and none of the healthy child controls) [46]. Fiorentino et al. analyzed postmortem brain tissues as well as duodenal tissue samples from ASD subjects revealing impaired gut barrier integrity as well as expression of genes associated with impaired blood–brain barrier integrity coupled with increased neuroinflammation [11]. Besides, intestinal barrier defects in the zonulin transgenic mouse model (ZTM) seem to have a critical effect on the intestinal microbiota composition, i.e., dysbiosis, as well as the immune system [73]. 

Latent pro-inflammatory status is created from uncontrolled trafficking of microbial products, which leads to skewed microbiota composition and immune profile. When challenged by an environmental trigger, the onset of overt inflammation can be produced with an increased risk of developing chronic disease [19,73]. Intestinal dysbiosis, including various metabolites, has been demonstrated both in animal models of ASD and in ASD children [74]. Intestinal dysbiosis could contribute to the low-grade systemic inflammatory state frequently reported in subjects with GI co-morbidities [74]. 

According to Hadjivassiliou et al., anti-neural TG6 antibodies are better serological markers of gluten intolerance in subjects with neurological symptoms than antibodies against intestinal TG2 [75]. In this review, occurrence of antibodies against neural anti-TG6 in ASD patients were found, which demonstrates a subgroup of ASD patients that responded to gluten with increased production of antibodies against native gluten and neural TG6 antibodies but not of typical celiac-specific antibodies. However, this production was not related to serological markers of increased IP [7].

A combination of increased IP and chronic inflammation may affect brain function and are associated with higher global psychiatric symptoms [12,76]. Disrupted homeostasis in the gut, i.e., dysbiosis, contributes to translocation of potential harmful pathogens via the enteric nervous system (ENS) and the vagal nerve from the gut to the brain [77]. The digestive tract interacts with the external environment, including the microbiota. The role of the intestines is to be selectively permeable for the intestinal luminal contents that will pass through the mucosal layer and affect the enteric ganglions before entering the systemic circulation where it will affect the central nervous system [15,78]. It is well established that the microbiome, i.e., the genetic material of the microbes, can significantly affect mental health, i.e., emotions and cognitive function. The HPA axis is sensitive to the microbiome, and research shows that modification of the intestinal microbiota by re-establishing intestinal eubiosis can alter HPA responsiveness [78,79].

Increased activity in the HPA axis has been observed in OCD subjects compared to controls [80]. Clinical evidence has been provided on alteration in microbiota diversity and complexity compared to controls, i.e., in ASD [81,82,83,84,85,86], ADHD [8,87], anxiety disorders, bipolar disorder [88,89], depression and schizophrenia [90,91,92,93,94], as well as in stress [95]. Microbiota affects intestinal integrity, and dysbiosis can lead to intestinal barrier dysfunction, which may trigger the immune system, affecting the nervous system [96]. While several studies have reported an altered microbiota in individuals with ASD, schizophrenia, or ADHD, there is limited or no evidence to demonstrate whether targeting the microbiota through probiotics or dietary interventions can improve symptoms in children. A limited number of studies investigating the therapeutic role of probiotics in individuals suffering from mood disorders, including stress, anxiety, bipolar disorder, and depression have been conducted to date. The microbiota–gut–brain axis might provide novel targets for the prevention and treatment of mental and neuropsychiatric disorders. However, further studies are required to substantiate the clinical use of probiotics and prebiotics.

According to this systematic review, pathologic findings such as inflammation of the GI system were common in subjects diagnosed with mental disorders, as well as reduced digestive enzyme activity and dysbiosis [6,7,8]. Furthermore, zonulin upregulation and subsequent increase in impaired barrier function in genetically susceptible individuals may lead to mental disorders. Therefore zonulin upregulation may be necessary but not sufficient to develop ASD, because other factors are likely to play a role [27]. Based on these observations, two subtypes of ASD could be considered, with different grades of inflammation, which might play a role in the etiopathogenesis of ASD and may explain the conflicting results of studies on dysbiosis and ASD [97,98]. The findings indicate that children diagnosed with ASD are more prone to inflammation in the gut when GI symptoms are included. All three studies on ASD and IP in the systematic review add to the knowledge of how impaired barrier function affects subjects diagnosed with ASD. Furthermore, to explain etiology in ADHD subjects and to provide additional information on potential therapies, further studies are needed on social communication skills and zonulin levels in patients diagnosed with ADHD. However, the findings on serum zonulin levels in OCD subjects revealed no difference compared to controls suggesting possible different mechanisms to be confirmed in further detailed studies. 

The results from the meta-analysis from this systematic review, revealed a significant difference between the patient and control groups when all three disorders, ADHD, ASD, and OCD were equated as “mental disorders”, which gives a more extensive comparison with increased sample size [97]. However, these findings need to be interpreted in terms of population and patient characteristics as well, in this systematic review. 

Many studies were excluded from this review. One of the main reasons was not fulfilling a priory eligibility criterion for biomarkers and reporting findings on altered IP from lactulose/mannitol testing. To be eligible for the current systematic review, the outcome needed to include the assessment of intrinsic biomarkers such as zonulin, which participate in the physiological regulation of intercellular tight junctions in the small intestine [99]. Dysregulation of the zonulin model may contribute to disease states that involve disordered intercellular communication, including developmental and intestinal disorders [5,6,7,9,17,27,29,100].

The findings of this systematic review and meta-analysis should be weighed against certain limitations. Eligibility criteria were limited to observational study type publications, and only five papers [5,6,7,8,9] met the inclusion criteria, including only three types of mental disorders, i.e., ADHD, ASD, and OCD. The findings from the meta-analysis need to be interpreted in terms of population and patient characteristics. However, and to the authors’ knowledge, there is no existing systematic review on the association between IP and mental disorders in children or adolescents. Therefore, the findings from this systematic review and meta-analysis are novel.

## 5. Conclusions

Three of the five studies included in this systematic review demonstrate an association between increased serum zonulin levels and impaired intestinal barrier function in mental disorders, i.e., in ADHD and ASD, and one study on increased plasma haptoglobin levels and impaired IP. One study reported increased serum claudin-5 levels and increased permeability of the blood–brain barrier in OCD. Mean zonulin levels, a marker of intestinal permeability, appear to be higher in children diagnosed with mental disorders compared to controls. However, the epidemiological evidence is limited, and methodological inconsistencies have been observed. Therefore, more studies are required in order to advance the field in this research area. Moreover, the efficacy of targeting the intestinal barrier in the management of neurological and behavioral aspects of mental disorders, i.e., central nervous systems disorders, has not yet been established and needs further investigation. Therefore, further studies are required in order to better understand the complex role of barrier function, i.e., intestinal and blood–brain barrier, and inflammation in the pathophysiology of mental and neurodevelopmental disorders.

## Figures and Tables

**Figure 1 nutrients-12-01982-f001:**
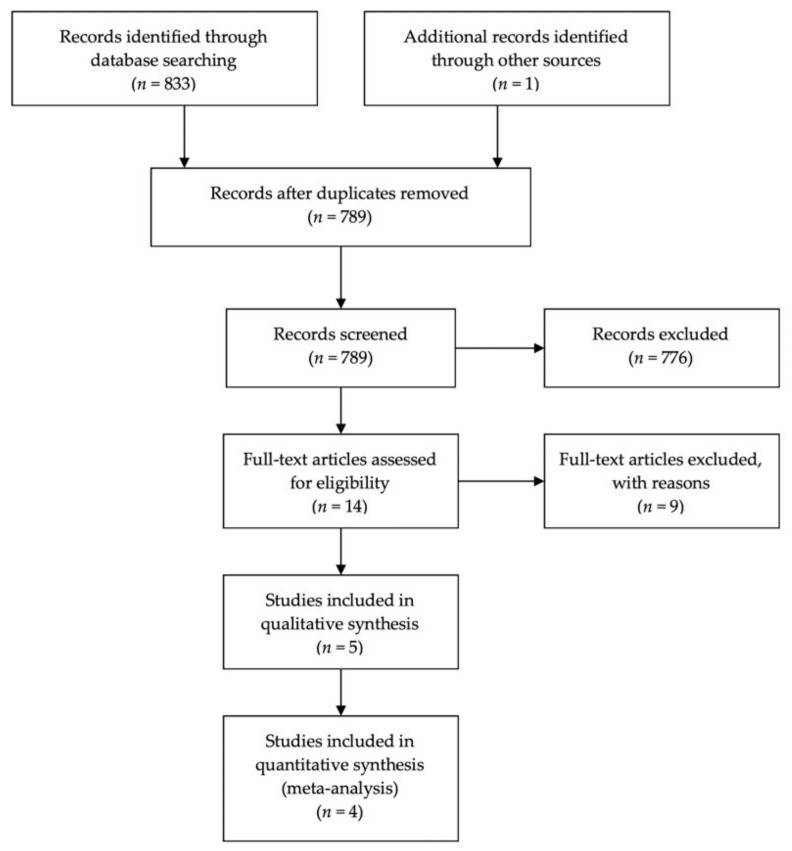
PRISMA flow diagram illustrating the results of the search, the process of screening, and the selection of the records for inclusion in the systematic review and meta-analysis.

**Figure 2 nutrients-12-01982-f002:**
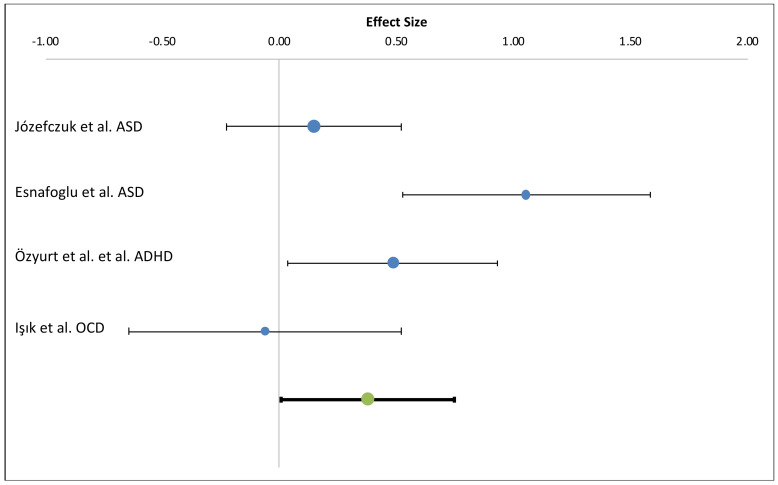
Forest plot showing the individual mean differences among cases and controls for each of the four studies and the combined estimate from a fixed-effect model (bottom, *p* = 0.001). The Effect estimates are presented as Cohen’s d with positive effect size indicating higher zonulin levels among cases compared to controls.

**Table 1 nutrients-12-01982-t001:** Inclusion criteria based on the PICO method.

Population	Children (0–18 years) diagnosed with one or more of the following according to validated criteria; anxiety disorders (AD), autism spectrum disorder (ASD), attention deficit disorder (ADD), attention deficit and hyperactivity disorder (ADHD), bipolar disorder (BD), major depressive disorder (MDD), manic-depressive disorder, obsessive–compulsive disorder (OCD), and schizophrenia. Mental disorders should be diagnosed by a standard procedure such as experts’ diagnoses involving dimensional and multi-informant assessments, including recognized and validated rating scales. To be eligible for inclusion, the study also needed to address intestinal permeability.
Exposures	Biomarkers (intrinsic or endotoxins) for intestinal permeability such as zonulin, diamine oxidase (DAO), lipopolysaccharide (LPS), and lipopolysaccharide binding protein (LBP) in individuals diagnosed with one or more of the mental illnesses according to the validated criteria.
Comparison	Biomarkers (intrinsic or endotoxins) for intestinal permeability such as zonulin, diamine oxidase (DAO), lipopolysaccharide (LPS), and lipopolysaccharide binding protein (LBP) in controls.
Outcome	Comparing intestinal permeability values (serum zonulin, stool zonulin, serum diamine oxidase, serum LPS, or serum LPB) between children diagnosed with mental, behavioral, or neurodevelopment disorders and controls.

**Table 2 nutrients-12-01982-t002:** Description of all excluded studies upon full-text review with reference I.D.

Reference ID	Study Characteristics	Reason for Exclusion
Calarger et al., 2019 [52]	Intervention study, major depressive disorder, children (age: 12–17), *n* = 41, analyzed urine; lactulose/mannitol recovery ratio.	Did not fulfill a priori inclusion criteria for biomarkers (zonulin, intestinal permeability).
Dalton et al., 2014 [53]	Intervention study, ASD, children (age: 10–14), *n* = 103, analyzed urine; lactulose/mannitol recovery ratio.	Did not fulfill a priori inclusion criteria for study type (observational) and participants (children 0–18 years).
Delaney et al., 2019 [54]	Observation cross-sectional study, Schizophrenia, children, adolescents, young adults (age: 8–35), *n* = 97, analyzed serum; 25 (OH) D, anti-LPS antibodies, CRP, IL-6 levels).	Did not fulfill a priori inclusion criteria for participants (children 0–18 years).
De Santis et al., 2019 [55]	Observational pilot study, ASD, children (age: 2–9), *n* = 110, analyzed mycotoxins.	Did not fulfill a priori inclusion criteria for biomarkers (zonulin, intestinal permeability).
Gabriele et al., 2016 [56]	Research study, ASD, children (age: 0–8), *n* = 53, analyzed urine; p-cresol levels.	Did not fulfill a priori inclusion criteria for study type, no controls.
Iovene et al., 2017 [57]	Intervention study, ASD, children (age: 3–9 approx.), *n* = 80, analyzed urine; lactulose/mannitol recovery ratio.	Did not fulfill a priori inclusion criteria for biomarkers (zonulin, intestinal permeability).
Jyonouchi et al., 2002 [58]	Intervention study, ASD, and DPI children, young adults, adults (age: 1–20), *n* = 209, analyzed; response to an elimination diet.	Did not fulfill a priori inclusion criteria for study type.
Lau et al., 2013 [59]	Cohort study, ASD, children (age: 4–12) *n* = 140, analyzed serum; IgG and IgA antibodies to gliadin.	Did not fulfill a priori inclusion criteria for biomarkers.
Pusponegoro et al., 2015 [60]	Observational cross-sectional study, ASD, children (age: 2–10), *n* = 268, analyzed urine; lactulose/mannitol recovery ratio.	Did not fulfill a priori inclusion criteria for study type.

**Table 3 nutrients-12-01982-t003:** Summary of the characteristics of the included case-control studies.

References	Country	Participants	Mean Age	Measurement/Diagnosis		
				ASD	ADHD	OCD
Özyurt et al., 2018 [5]	Turkey	Total Blood (81). 40 ADHD/41 HC	ADHD: 7.9/HC: 7.8	N/A	Blood: Serum Zonulin/DuPaul ADHD-RS-IV Inventory	N/A
Esnafoglu et al., 2017 [6]	Turkey	Total Blood (65). 32 ASD/33 HC	ASD: 7.5/HC: 7.0	Blood: Serum Zonulin and BMI/CARS	N/A	N/A
Józefczuk et al., 2017 [7]	Poland	Total Blood (121). 75 ASD/46 HC	ASD: 8.1/HC: N/A	Blood: Serum Zonulin and CSA, AGA, I-FABP, /ADI-R, ADOS Scale	N/A	N/A
Rose et al., 2018 [8]	U.S.A. California	Total Blood (87). 46 ASD (NoGI: 26, GI: 20), 41 TD (NoGI: 35, GI:6; Total Stool (91). 50 ASD (NoGI: 29, GI: 21), 41 TD (NoGI: 34, GI: 7)	Blood: ASD- NoGI: 7.8/TD-NoGI: 6.8; ASD-GI: 5.7/TD-GI: 5.2; Stool: ASD- NoGI: 7.8/TD-NoGI: 7.1; ASD-GI: 6.6/TD-GI: 5.1	Blood: Plasma Haptoglobin and Stool: Microbiome/ADI-R, ADOS	N/A	N/A
Işık et al., 2020 [9]	Turkey	Total Blood (48). 24 OCD/24 HC	OCD: 14.3/HC: 13.7	N/A	N/A	Blood: Serum Zonulin and serum Claudin-5 and BMI/K-SADS-PL, DSM-5 and CY-BOCS, M.O.C.I., RCADS-CV

N/A: Not available, ASD: autism spectrum disorders, ADHD: attention deficiency hyperactivity disorder, OCD: obsessive–compulsive disorder, HC: healthy controls, CSA: celiac-specific antibodies, AGA: anti-gliadin antibodies. ADI-R: Autism Diagnostic Interview-Revised, ADOS: Autism Diagnostic Observation Schedule, DuPaul ADHD-RS-IV Inventory: DuPaul attention deficit hyperactivity disorder rating inventory, CARS: Childhood Autism Rating Scale, K-SADS-PL: Kiddie Schedule for Affective Disorders and Schizophrenia, DSM-5: DSM-5: Diagnostic and Statistical Manual of Mental Disorders, Fifth Edition, CY-BOCS: Children’s Yale–Brown Obsessive–Compulsive Scale, MOCI: Maudsley Obsessive–Compulsive Inventory, RCADS-CV: Revised Child Anxiety and Depression Scales–Child Version, BMI: Body Mass Index, I-FABP: Intestinal fatty acid binding protein. ASD—GI: with ASD and GI symptoms of irregular bowel habits, ASD—NoGI: with ASD but without current or previous GI symptoms, TD-GI: typically developing children with GI symptoms, TD-NoGI: typically developing children but without current or previous GI symptoms.

**Table 4 nutrients-12-01982-t004:** Summary of main findings.

First Author, Year of Publication, Country	Participants Included in the Study Based on Serum Zonulin and Plasma Haptoglobin	Main Study Findings	Authors Conclusions
Özyurt et al., 2018, Turkey [5]	81	Children with ADHD had significantly elevated levels of zonulin compared to controls. Children with hyperactive/impulsive presentations had significantly elevated zonulin compared to other presentations. ADHD symptoms and social communication problems correlated significantly with zonulin levels, and hyperactive/impulsive and social communication symptoms were important predictors of zonulin levels.	“Regardless of its limitations, the results of our study suggest that zonulin levels may be elevated in children with ADHD (especially the hyperactive/impulsive presentation) and that this elevation correlated with social deficits. Symptoms of hyperactivity/impulsivity and social deficits independently predict zonulin levels in children with ADHD although the changes in adjusted R2 suggest that the majority of the predictive value lies with symptoms of hyperactivity/impulsivity.”
Esnafoglu et al., 2017, Turkey [6]	65	There was an increase in serum zonulin levels in the group with ASD compared with the healthy control group. Additionally, for all subjects, there was a positive correlation identified between the CARS score, indicating severity of autism, and zonulin.	“Increased zonulin levels in patients with ASD may play a role in the development of ASD symptoms. However, zonulin upregulation and subsequent increase in intestinal permeability may be necessary but not sufficient to develop ASD, because other factors are likely at play.”
Józefczuk et al., 2017, Poland [7]	121	Concentrations of zonulin were the highest in the youngest children (5 years). The mean level of zonulin in this group was significantly higher compared with patients aged 6–11 years. The occurrence of anti-TG6 antibodies in ASD patients with normal mucosa was not associated with CD.	“There is a subgroup of ASD patients whichrespond to gluten with increased production of antibodies against native gluten and neural TG6, but not of typical celiac-specific antibodies, and this production is not related to serological markers of an impaired intestinal barrier.”
Rose et al., 2018, California [8]	87	Children with ASD who experience GI symptoms have an imbalance in their immune response, possibly influenced by or influencing metagenomic changes, and may have a propensity to impaired gut barrier function, which may contribute to their symptoms and clinical outcome.	“We found several differences when comparing children with ASD who exhibit GI symptoms vs. those that did not. The most notable of these was the reduced regulatory TGFb1 response of the ASDGI groups following stimulation. We also noted an increase in the production of cytokines linked to mucosal inflammation after TLR-4 stimulation in children with ASDGI symptoms relative to children with ASDNoGI. Our analysis of the microbiome underscores the relationship between the presence of GI symptoms and the host microflora and suggest a possible role of dysbiosis in the co-morbidity of GI issues in ASD”
Işık et al., 2020, Turkey [9]	48	There was an increase in serum claudin-5 levels in the group with OCD compared with the healthy control group. There was no significant difference between the study and control group in the serum zonulin concentrations.	“Regardless of the limitations, taken together with our results, dysregulation of the BBB, especially claudin-5, may be involved in the etiology of OCD. Further detailed and more comprehensive studies designed on a longitudinal basis are greatly needed to find out exactly whether increased claudin- 5 levels are the cause or consequence of the disease process in OCD”

ADHD: attention deficit hyperactivity disorder. CARS: childhood autism rating scale. ASD: autism spectrum disorders.TG6: transglutaminase 6. IgA: immunoglobulin A. IgG: immunoglobulin G. CD: celiac disease. GI: gastrointestinal. OCD: obsessive–compulsive disorder. BBB: blood–brain barrier.

**Table 5 nutrients-12-01982-t005:** Serum zonulin levels (ng/mL) in patients and control groups.

Study	Serum Zonulin Levels (ng/mL) in Patients and Control Groups						
	Patient Group			Control Group			
	*n*	Mean Zonulin	S.D.	*n*	Mean Zonulin	SD	*p*-Value
Özyurt et al., ADHD (*n*)/(ng/mL), mean ± SD	40	105.36	98.38	41	63.34	73.4	0.031	
Esnafoglu et al., ASD (*n*)/(ng/mL), mean ± SD	32	122.3	98.46	33	41.89	45.83	<0.001	
Józefczuk et al., ASD (*n*)/(ng/mL), mean ± SD	75	17.2	15.7	46	15.3	5.9	>0.05	
Işık et al., OCD (*n*)/(ng/mL), mean ± SD	24	98.93	83.06	24	103.7	73.53	0.834	

**Table 6 nutrients-12-01982-t006:** Quality assessment of the included case-control studies by the N.O.S. scale.

Study	Selection of Controls	Definition of Controls	Non- Response Rate	Final Score
Özyurt et al., ADHD [5]	0	*	*	8/9 (strong)
Esnafoglu et al., ASD [6]	0	*	*	8/9 (strong)
Józefczuk et al., ASD [7]	0	0	0	6/9 (moderate)
Rose et al., ASD [8]	*	*	0	8/9 (strong)
Işık et al., OCD [9]	0	*	0	7/9 (strong)

One “*” means one point, “0” means no point.

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
