# Peer review of "Zonulin-Dependent Intestinal Permeability in Children Diagnosed with Mental Disorders: A Systematic Review and Meta-Analysis"

_nutrients, 2020, doi:10.3390/nu12071982_

Round 1
Reviewer 1 Report
This is a very comprehensive review of the existing literature and carefully outlines the state of the field. Because of the rigorous standards of the inclusion criteria, however, the number of papers reviewed is quite low. Is there any way to speak to the findings of the other papers which were acceptable but not included because of other reasons, like biomarkers, for example?
Author Response
Thank you for a very good comment which was very well appreciated by the authors. We have not conducted a brief summary of the main findings from the papers which were excluded from the SR by reasons. The summary has been added to the text between lines 425 and 448. The findings from five of the nine excluded studies add to the findings of this SR. In addition, English language has been revised and typographical errors corrected. These were performed using the "tracked changes" function.
Best Regards,
Birna
Reviewer 2 Report
The authors describe in a very detailed way about the selection process of articles. Everything seems quite sound and well done until the actual meta-analysis. Search terms, inclusion criteria, bias check-up make sense, but the authors seem to lack expertise in performing a meta-analysis per se. Data about OR/RR including CI with I² estimate together with visualization by forest-plots would be appropriate to get a feeling for the extracted data and to be able to draw any conclusions.
The authors are encouraged to do a real meta-analysis, as plotting such data by means of bar charts is not common.
Please also check the article again for language and typos.
Author Response
Thank you very much for your comments which were very well appreciated. The authors have now completed the revision for language and typos as well as for the meta-analysis, and made the following changes, all performed using the Tracked Changes funtion:
In chapter 2.5. "Meta-analysis" (line 223) the t-test was deleted and we applied a fixed-effects model using Cohens-d to calculate the effect size.
In chapter 3.5. "Meta-analysis" (lines 335-341) a real meta-analysis is explained and publication bias assessed. Between lines 346 and 354 a new Figure was added and the former deleted, i.e., Figure 2 is now a Forest plot instead of a Bar chart.
Other corrections were only related to language and typos and all performed using the "Track Changes" function.
Best Regards,
Birna